# Brain-Derived Neurotrophic Factor in Central Nervous System Myelination: A New Mechanism to Promote Myelin Plasticity and Repair

**DOI:** 10.3390/ijms19124131

**Published:** 2018-12-19

**Authors:** Jessica L. Fletcher, Simon S. Murray, Junhua Xiao

**Affiliations:** Department of Anatomy and Neuroscience, School of Biomedical Sciences, Faculty of Medicine, Dentistry and Health Sciences, The University of Melbourne, Parkville, 3010, VIC, Australia; ssmurray@unimelb.edu.au

**Keywords:** oligodendrocyte, BDNF, TrkB, myelination, neurotrophin

## Abstract

Brain-derived neurotrophic factor (BDNF) plays vitally important roles in neural development and plasticity in both health and disease. Recent studies using mutant mice to selectively manipulate BDNF signalling in desired cell types, in combination with animal models of demyelinating disease, have demonstrated that BDNF not only potentiates normal central nervous system myelination in development but enhances recovery after myelin injury. However, the precise mechanisms by which BDNF enhances myelination in development and repair are unclear. Here, we review some of the recent progress made in understanding the influence BDNF exerts upon the myelinating process during development and after injury, and discuss the cellular and molecular mechanisms underlying its effects. In doing so, we raise new questions for future research.

## 1. Introduction

### 1.1. Central Nervous System Myelination

Myelin has had an evolutionarily important role in enabling rapid nerve impulse conduction in both the peripheral and central nervous systems, and provides trophic and metabolic support to the axons it ensheaths [1,2,3]. There is growing consensus that myelin is not only critical for normal motor and sensory function, but that it also participates in higher order functions such as cognition, learning and memory [4,5,6,7]. Central nervous system (CNS) myelination is a complex sequence of cellular events involving proliferation and migration of oligodendrocyte progenitor cells (OPCs), recognition of target axons and axon-glia signalling, differentiation of OPCs into mature myelinating oligodendrocytes (or alternatively, apoptosis), membrane outgrowth and axonal wrapping, myelin elongation and compaction, and node formation. This myelinating process is tightly controlled by key transcription factors [8,9,10,11] that are responsive to a range of extracellular cues, including molecules and ligands secreted by or expressed on the surface of axons [10,11]. Significant evidence implicates the neurotrophin brain-derived neurotrophic factor (BDNF) as a key pro-myelinating molecule. Precise understanding of the cellular and molecular mechanisms by which BDNF engages intracellular signalling cascades critical to myelination has the potential to inform novel therapeutic approaches to promote remyelination.

### 1.2. Early Discoveries of the Role of BDNF in Central Nervous System (CNS) Myelination

BDNF is a member of the neurotrophin family and is the most widely expressed neurotrophin in the mammalian brain [12]. BDNF is initially synthesised as a precursor protein, proBDNF, that is subsequently cleaved to release mature BDNF which then exerts biological activities via two receptors. These receptors are tropomyosin-related kinase (Trk) B and the structurally unrelated p75 neurotrophin receptor (p75^NTR^) [12,13]. It is well established that both neurons and glia, including the myelin-forming oligodendrocyte, express both BDNF and its receptors. In addition, there are truncated isoforms of TrkB which are predominantly expressed by astrocytes that can evoke calcium transients [14] rather than canonical intracellular kinase pathways. Complexity in the combinations of ligand-receptor interactions that are possible, in addition to the cell-specific function these interactions elicit, emphasises the challenge that exists in understanding how BDNF regulates CNS myelination.

BDNF is historically known for its critical roles in neuronal function [13]. Investigation of its role in glial cell functions such as myelination began only two decades ago and has rapidly advanced following the generation of the BDNF knockout (KO) mouse in the mid-1990s [15]. BDNF KO mice exhibited significant decreases in the expression of a key myelin protein, myelin basic protein (MBP), as well as reduced mRNA transcripts of MBP and proteolipid protein (PLP) in the hippocampus and cortex [16]. Importantly, these molecular changes were reflected in ultrastructural changes to myelin. In one study involving 2-week old BDNF KO mice, the proportion of myelinated axons in the optic nerve was reduced by 50% compared to wildtype littermate mice, but there was no change to the number of retinal ganglion cells, or to the size or organisation of the retinal layers [17]. This suggests an effect of BDNF specific to myelination rather than neuronal development. This reduction in myelination persisted until the third postnatal week and was accompanied by a significant reduction in optic nerve axon diameter, indicative of a greater number of small unmyelinated axons in BDNF KO mice [17]. This may reflect a potential issue in the mechanisms that control the initiation of myelination, given that the retinal ganglia population size was normal. BDNF KO mice have poor health that results in premature death [16,17], giving them limited utility for the examination of myelin development into adulthood. It remains unclear whether myelin deficits in BDNF KO mice are directly due to BDNF loss or are secondary to global developmental delay.

The shortened lifespan of the BDNF KO mouse have led to the analysis of the BDNF heterozygous (HET) mouse to interrogate the influence of BDNF on CNS myelin development. BDNF HET mice have a normal lifespan but exhibit a 40% reduction in BDNF expression [18,19]. BDNF HET mice also display significant reductions in the expression of myelin proteins (including MBP, PLP, myelin associated glycoprotein (MAG) and myelin oligodendrocyte glycoprotein (MOG)) in the forebrain, corpus callosum, spinal cord and optic nerves [18,19,20]. This is indicative that endogenous BDNF potentiates normal CNS myelination during development in vivo. These findings are strongly supported by in vitro work, where exogenous BDNF significantly enhances myelin formation in myelinating co-cultures using dorsal root ganglion neurons and OPCs. BDNF treatment has resulted in significantly more MBP+ myelinated axonal segments than basal conditions [19]. Collectively, these in vivo and in vitro findings demonstrate that BDNF is a strong pro-myelinating factor in the CNS.

Understanding the pro-myelinating influence BDNF exerts upon CNS myelination requires knowledge of the molecular signalling cascades it elicits in both the oligodendrocyte and the axon (Figure 1a). BDNF signals through both p75^NTR^ and TrkB. The p75^NTR^ KO mouse exhibits a normal lifespan and despite severe sensory neuron loss [21] and myelin deficits in the peripheral nervous system [22], there are no reports suggesting p75^NTR^ KO results in abnormal CNS myelination. Examination of myelin protein expression in p75^NTR^ KO mouse brains has revealed no apparent changes compared to wildtype mice [23]. This indicates that p75^NTR^ does not directly mediate the pro-myelinating effect of BDNF. Indeed, BDNF still exerts its pro-myelinating effect when added to myelinating co-cultures using either OPCs or axons from p75^NTR^ KO mice [19,24]. This strongly implicates TrkB as the critical receptor required for the influence BDNF exerts upon CNS myelination. Supporting this, pharmacological inhibition of Trk receptors on oligodendroglia in vitro block the pro-myelinating influence of BDNF [19,24]. The global TrkB KO is neonatal lethal [25], limiting its utility in understanding BDNF-induced CNS myelination, and necessitating a conditional KO strategy to determine the relative contribution of oligodendroglial or neuronal TrkB activation on CNS myelination in vivo.

## 2. BDNF Promotes Developmental Myelination *via* TrkB

### 2.1. The Role of BDNF in Oligodendroglial Proliferation During Development

It is clear from BDNF HET mice studies that BDNF promotes developmental myelination in vivo [18,19,20,26]. However, whether this is through a direct effect upon oligodendroglial survival or proliferation remains controversial. This is due to the heterogenous responses to BDNF from oligodendrocytes sourced from distinct CNS regions have been observed [26,27,28]. Initial in vitro studies have found BDNF promoted oligodendroglial proliferation in OPCs derived from the basal forebrain, but not in OPCs from the optic nerve [27,28]. Variation in levels of TrkB expression amongst oligodendroglia likely accounts for these differing observations. Both full-length and truncated TrkB isoforms are readily detectable in basal forebrain oligodendrocytes, while lower levels of full-length TrkB are expressed in oligodendrocytes from the cortex [19,26,29]. This suggests that BDNF potentially exerts diverse influences upon oligodendroglia, which are highly contextual to specific CNS regions.

Such heterogenous and regionally distinct effects of BDNF on oligodendroglial proliferation, survival and differentiation are also supported by in vivo findings. A recent systemic spatio-temporal analysis of oligodendroglial populations in BDNF HET mice has demonstrated a regionally distinct and transient reduction in oligodendroglia in response to BDNF haploinsufficiency [30]. Across multiple grey and white matter regions, the most robust change in oligodendroglia was a 50% reduction in the optic nerve at postnatal day (P) 9. Smaller reductions were observed in the spinal cord, but there were no changes in oligodendroglial densities in the corpus callosum, cerebral cortex or optic nerve. Notably, these reductions were only transient. Oligodendroglial density normalised to wildtype levels in the above regions by P30 [30]. These findings suggest that the influence BDNF haploinsufficiency exerts on oligodendroglial survival or proliferation in vivo is modest and transient. BDNF HET mice continue to express both TrkB and BDNF, albeit BDNF levels are approximately half. The modest reductions seen in oligodendroglia due to BDNF haploinsufficiency [18,30] may be reflective of this persistent low level of BDNF signalling. It may alternatively reflect signalling redundancy within the myelination program that compensates for the reduced BDNF signal. Importantly, this could effectively conceal more substantial effects of BDNF upon oligodendroglial populations. Alternate strategies to conditionally and specifically delete TrkB from OPCs, ideally at the time of their specification as can be achieved using the Olig2 promoter [31], are required in order to definitively assess the effect of BDNF on OPC survival and proliferation (Figure 1b).

### 2.2. The Role of BDNF in Developmental Myelinogenesis

Despite limited evidence that BDNF robustly influences oligodendroglial proliferation, it is widely accepted that it signals through oligodendrocyte-expressed TrkB to promote myelin synthesis in vitro [19,26,28,32] and in vivo [33] (Figure 1d). Indeed, generation of conditional TrkB KO mice in which TrkB was deleted from oligodendrocytes under the control of the MBP promoter [33] has been critical in defining the influence that BDNF-TrkB signalling exerted upon the myelinating process. These mice have exhibited the normal density and size of myelinated axons during development. However, the myelin produced was significantly thinner [33]. These findings indicate oligodendroglial BDNF-TrkB signalling has no effect on the initial contact of the oligodendrocyte with the axon. Instead, there is a specific effect on the rate of ensheathment, with normalisation of myelin protein expression by adulthood [33]. Intriguingly, myelin abnormalities in MBP conditional TrkB KO mice [33] are discordant with the phenotype observed in BDNF HET mice [19] and suggest TrkB signalling in additional cell type(s) influences the early events of myelination, opening new areas of investigation. This possibility highlights that the molecular signals regulating initial contact between oligodendrocytes and axons are distinct from the signals controlling either production of myelin constituents or the membrane elongation and wrapping that comprise ensheathment. The density of maturing oligodendrocytes in these conditional TrkB KO mice was normal [33]. This supports the findings from the BDNF HET mice that BDNF-TrkB signaling has little effect on oligodendroglial differentiation or survival in vivo. However, one curious observation from the conditional TrkB KO mice is that targeted TrkB deletion in MBP-expressing oligodendrocytes increased OPC density [33]. As oligodendroglia do not commence MBP expression until they are post-mitotic, recombination in OPCs is extremely low [33], suggesting this proliferative effect is indirect of BDNF-TrkB signalling.

Pharmacological and genetic analyses have interrogated the signalling elements responsible for the pro-myelinogenic effect that oligodendroglial TrkB exerts. It is now well established that BDNF-TrkB activates the mitogen-activated protein kinase (MAPK)/Erk pathways to promote oligodendrocyte differentiation and myelination in vitro [26,28,32,34] and in vivo [35,36,37,38] (Figure 1d). It is noteworthy that mice with oligodendrocyte-specific loss of TrkB or Erk1/2 activation share a similar hypomyelinating phenotype: significantly thinner myelin but a normal number of maturing oligodendrocytes. The critical difference between these two types of mice is that the myelin deficit in the oligodendrocyte TrkB KO is resolved by adulthood [33], whereas in the Erk1/2 conditional KO mice, the phenotype persists [36]. As other oligodendrocyte-expressed growth factor receptors also activate the Erk1/2 pathway including FGFR2 [39], integrin receptors [40] and Tyro3 [41], it is likely that other growth factors ultimately compensate for the loss of BDNF-TrkB signalling to normalise myelination. Collectively these data suggest that oligodendrocyte BDNF-TrkB signalling activates Erk1/2 to drive formation of the myelin sheath. 

It remains unanswered how TrkB activation promotes myelinogenesis via signalling to Erk1/2. It could be indirect, as Erk1/2 are well known to cross-talk with other signalling pathways, such as those involving Fyn, Akt and GSK3 (all of which have been independently implicated in regulating oligodendrocyte myelination) [42,43,44,45,46,47]. Alternatively, Erk1/2 are also well known as transcriptional activators capable of phosphorylating and activating specific transcription factors to alter gene expression [48]. Genetic studies have identified a range of transcription factors that control the myelin program [10,49]. However, whether Erk1/2 associates with and activates any of these remains to be seen (Figure 1d). Also unresolved is the emergence of myelin plasticity, and how experimental evidence now informs us that skill acquisition [50] and electrical activity [5] can modulate myelination. The influence Erk1/2 exerts upon myelination during development is clear, but whether it influences these finer adaptive aspects of myelination is unknown.

### 2.3. The Dual Role of BDNF in Activity-Dependent and -Independent Myelination?

There is growing consensus that CNS myelination occurs by two modes: the first is independent of neuronal activity also known as innate or intrinsic myelination, and is the historic perspective under which CNS myelination has been understood [27,51,52,53,54,55]. The second mode is activity-dependent myelination, in which oligodendrocyte proliferation, differentiation and their capacity to myelinate axons in vitro and in vivo is dependent on neural activity [5,56,57,58,59,60]. While knowledge that axonal activity influences the myelinating process has been accepted for over 20 years [56], understanding of the molecular signals coordinating the oligodendroglial response to axonal firing is incomplete. Recent studies [51,61] have implicated a potential role for BDNF in activity-dependent myelination (Figure 1c).

The expression of BDNF itself is regulated by neural activity [62,63] (Figure 1a), including sensory and environmental experiences [64,65]. Neuronal activity regulates both the transcription of the BDNF gene and the transport of BDNF mRNA and protein into dendrites [66], possibly involving activation of presynaptic NMDA glutamate receptors [67]. Trafficking of TrkB, including its cell surface expression and ligand-induced endocytosis, may also be regulated in an activity-dependent manner [62]. Extensive studies have established that BDNF regulates activity-dependent synaptic plasticity [13,62]. Given the role of neural activity in BDNF release, and the established roles of BDNF-TrkB signalling in CNS myelination [16,17,18,19,20,33,68], it is reasonable to speculate that BDNF participates in activity-dependent myelination (Figure 1c), including experience-dependent adaptive myelination.

Clearly, BDNF signalling via oligodendroglial TrkB can influence the extent of myelin protein synthesis in the absence of axons [24,26]. That said, emerging evidence suggests BDNF is involved in the two modes of CNS myelination simultaneously [61]. BDNF is released in response to neuronal activity [66,69,70], and elevation in BDNF levels may initiate a cascade of glutamatergic neurotransmission [67,71,72,73] that activates N-methyl-D-aspartate (NMDA) and α-amino-3-hydroxy-5-methyl-4-isoxazolepropionic acid (AMPA) receptors on OPCs [61,74] (Figure 1d). Glutamate receptor activation may increase OPC energy supply, promoting their survival, proliferation or differentiation [75,76]. Conceivably, both innate and activity-dependent myelination may occur within the same or different neural circuits. It is plausible that an activity-independent intrinsic pathway may pre-establish an initial pattern of oligodendrocyte myelination which is then controlled and modified by activity-dependent cues such as BDNF to meet fine-tuned neuronal circuit development and functions. The potential dual roles of BDNF in the two modes of CNS myelination could have important implications in myelin plasticity under steady-state conditions, and neural pathology including psychiatric disorders [4,77,78].

## 3. BDNF-TrkB Signalling to Stimulate Myelin Regeneration Following Demyelination

Knowledge that BDNF promotes CNS myelination through TrkB signalling suggests it could be targeted as a therapeutic for demyelinating conditions, including stroke, traumatic injury and multiple sclerosis (MS). Direct infusion or cell-based gene therapy to overexpress recombinant BDNF in stroke and spinal cord injury, significantly increased OPC numbers and expression of myelin proteins MBP, MOG and PLP [79,80]. The enhanced myelination effect in neurotrauma is consistent with the role BDNF has in developmental myelination. However, the poor pharmaco-kinetics of BDNF, including its 28kDa molecular size, rapid half-life within blood circulation [81], and capacity to mediate opposing effects through its dual receptors TrkB and p75^NTR^ [82], make BDNF unfavourable for clinical implementation. Instead, small molecular weight compounds with greater stability that functionally mimic the effects of BDNF through p75^NTR^ or TrkB independently have been developed [83,84,85,86,87,88]. This offers potential to selectively elicit specific biological outcomes of BDNF function, including myelin formation. 

### 3.1. Use of TrkB Agonists to Promote Myelin Repair 

Tricyclic dimeric peptide-6 (TDP6) and 7,8-dihyrdroxyflavone (DHF) are two TrkB agonists shown to functionally mimic BDNF by phosphorylating TrkB and preventing apoptosis in neuronal survival assays in vitro [84,86]. Both have been used in animal models of the auto-immune demyelinating disease MS [89,90], a disease in which remyelination eventually fails [91,92]. Importantly, for a TrkB-targeted remyelination strategy there is evidence that TrkB expression is elevated in brain lesions of MS patients [93]. DHF has been used prophylactically from the day of disease induction in the experimental autoimmune encephalomyelitis (EAE) model of MS [90]. Consequently, its ability to specifically and directly promote myelin regeneration through oligodendroglial TrkB is unclear. DHF may also act as an anti-oxidant [94] and may ameliorate the inflammation that potentiates EAE. In support of DHF having a functional pro-myelination effect, increased expression of myelin proteins, as well as oligodendrocyte and OPC numbers in response to daily DHF treatment, have been reported [90]. While modelling the immune component of MS is a strength of EAE, it is a continuing challenge to accurately assess de- and subsequent remyelination [95]. Since in this case DHF treatment commenced on the day of EAE induction [90], it is equally possible that increases in myelin protein and oligodendroglia are indicative of myelin preservation rather than remyelination. 

TDP6 is a structural-mimic of BDNF that promotes oligodendrocyte myelination in vitro, with this effect abrogated upon genetic deletion of oligodendroglial TrkB [68]. These findings have been extended in vivo with TDP6 enhancing remyelination through activation of oligodendroglial TrkB in the cuprizone model of demyelination [89]. MS involves complex interactions between the immune system and CNS, and currently there is no animal model that faithfully imitates all aspects of MS pathology. While the cuprizone model does not have an autoimmune inflammatory component, it is a toxin-induced model that follows a stereotypical pattern of focal demyelination, followed by reproducible remyelination upon cuprizone withdrawal [96,97]. It is a powerful and commonly used experimental model to examine the biology of myelin regeneration and assess the remyelinating capacity of therapeutic candidates in the mammalian CNS. 

Direct infusion of TDP6 into the brains of cuprizone demyelinated mice resulted in an increased TrkB phosphorylation on oligodendrocytes, as well as the proportion of myelinated axons, thicker myelin sheaths and increased numbers of post-mitotic oligodendrocytes [89]. Importantly, TDP6 treatment has been directly compared to infusion of exogenous recombinant BDNF. BDNF treatment demonstrated an increase in myelin sheath thickness, but exerted no effect on the number of remyelinated axons or oligodendrocyte populations. These inconsistencies may relate to differences in molecular size, dose and stability of TDP6 compared to BDNF [89]. However, they could also be indicative of possible TrkB-specific actions of BDNF in oligodendroglial proliferation and survival. TDP6 is selective for TrkB and there is no evidence it engages p75^NTR^ [84,89]. Comparing the action of TDP6 and attenuated response to BDNF on oligodendrocyte numbers during remyelination, it is tempting to speculate that neurotrophin co-receptor complexes [33,98] may be formed that modulate the influence of BDNF on oligodendroglial sub-populations.

### 3.2. How Does TrkB Activation Promote Remyelination?

#### 3.2.1. A Role for TrkB Activation in Neural Precursor Cell Recruitment During Myelin Repair?

Increased myelin sheath thickness in response to TrkB activation by TDP6 is consistent with the activation of MAPK/Erk signalling pathways within oligodendrocytes in the context of both adult myelin maintenance and repair [35,99,100], and the effects BDNF demonstrates in developmental myelinogenesis [19]. Increased myelin sheath thickness by TDP6 required oligodendroglial expression of TrkB and was not observed when TDP6 was administered to mice with TrkB deleted from maturing oligodendrocytes [89]. In the cuprizone model, myelin produced by oligodendrocytes derived from the sub-ventricular zone (SVZ) population of neural precursor cells (NPCs) during remyelination is thicker [101]. Comparing these findings prompts several questions: (1) what is the origin of oligodendrocytes producing thicker myelin in response to TrkB activation? (2) is it possible for BDNF to influence NPC-derived OPC migration? and (3) can BDNF-TrkB signalling influence the conversion of NPCs to OPCs?

In TDP6-treated mice, analysis of myelin sheath thickness after remyelination was restricted to the midline of the caudal corpus callosum [89]. This is a region where NPC-derived OPCs and oligodendrocytes are in lower densities, but immature pre-myelinating oligodendrocytes derived from NPCs are in their highest numbers [101]. Curiously, the midline of the caudal corpus callosum is also where the contribution of NPC-derived oligodendroglia to remyelination is lowest, with thin myelin sheaths following six weeks remyelination [101]. It is unlikely the origin of oligodendrocytes producing thicker myelin sheaths in response to TrkB activation is the SVZ. This phenomenon is more likely a direct consequence of BDNF-TrkB-Erk signalling on parenchymal oligodendroglia resident in the caudal corpus callosum. 

While it is unlikely that thicker myelin produced in response to TrkB activation is from NPC-derived oligodendrocytes, whether BDNF-TrkB can influence NPC-derived OPC migration or fate-commitment to the oligodendrocyte lineage in remyelination remains unanswered. The five-week difference in the duration of remyelination examined in these two studies [89,101] is an important caveat to any conclusion that BDNF-TrkB cannot have these effects during remyelination. It is well-established that BDNF influences proliferation [102], migration [103,104], and neuronal fate-commitment of NPCs in multiple neurogenic zones [102,105]. However, there are only limited in vitro data suggesting BDNF through TrkB alters differentiation of NPCs towards the oligodendrocyte lineage [106], and most in vivo studies examining the role of BDNF on glial fate-commitment have focused on astrocytes as the outcome using GFAP as a marker [102,105]. Notably, none of these studies have been performed in de- or remyelinating conditions. To definitively answer if BDNF-TrkB influences NPC-derived OPC migration or NPC fate-commitment towards oligodendroglia during remyelination, comprehensive combined gain- and loss of function lineage tracing experiments in combination with mutant mice selectively deleting TrkB in NPCs or OPCs in a demyelinating model are required. Critically, functional BDNF-mimetics could be used to elucidate potential TrkB-specific effects.

#### 3.2.2. Does TrkB Activation Promote Remyelination *via* OPC Proliferation After Injury?

Increased post-mitotic oligodendrocyte density following TrkB activation during remyelination [89] is likely a consequence of TrkB-specific effects on parenchymal OPC proliferation and survival, prior to differentiation. BDNF-TrkB signalling on oligodendrocytes predominantly promotes myelin sheath synthesis in development [33,107], so it is reasonable to argue the increase in maturing oligodendrocytes in response to TrkB activation is purely due to accelerated differentiation. However, this does not consider that to have more post-mitotic oligodendrocytes there must first be more OPCs, either through greater proliferation or enhanced survival.

Evidence for BDNF-TrkB involvement in regulating OPC proliferation during remyelination comes from cuprizone studies in BDNF HET mice [108,109]. Demyelination inherently stimulates OPC proliferation [91,96,108] and in BDNF HET mice OPC proliferation in response to four- or five-weeks demyelination was diminished [108,109]. This is indicative that generation of replacement OPCs in remyelination is sensitive to microenvironmental BDNF levels. However, there were no increases in oligodendroglia when cuprizone demyelinated animals were treated with exogenous BDNF, or in the proliferative fraction of OPCs in mice treated with TDP6 [89]. These are increases that might be expected if BDNF-TrkB acts as a positive signal for OPC proliferation. Further obfuscating a potential role for BDNF-TrkB on OPC proliferation is that at six-weeks demyelination, deletion of oligodendroglial TrkB had no influence on OPC or post-mitotic oligodendrocyte densities compared to wildtype mice [89]. This may relate to the remyelination that can commence during the fifth to sixth week of cuprizone feeding [97]. However, overall it is difficult to reconcile the specific role or direction BDNF-TrkB signalling has on OPC proliferation; additional data show the post-mitotic oligodendrocyte population returns to healthy levels after four weeks recovery in BDNF HET mice [109], and in development, either oligodendroglial TrkB [33] or neuronal BDNF [107] deletion results in OPC hyperproliferation. It is also unclear whether the possible role of BDNF on OPC proliferation is specific to the acute demyelinating context. Alternatively, OPC proliferation could be independent of oligodendroglial TrkB, and could instead be a response to neuronal or astrocytic signals that become dynamic after injury. Indeed, both neurons and astrocytes secrete BDNF and express TrkB receptors, and BDNF release from these cell-types would be diminished in BDNF HET mice. Determination of whether BDNF-TrkB signalling has a direct effect on OPC proliferation following injury could be achieved using conditional KO mice with pharmacological manipulation, combined with birth-dating protocols to label dividing cells at the time of cuprizone-feed removal and onset of remyelination.

Activating TrkB using TrkB agonists [89,90] or by elevating BDNF levels [79,80,110] promotes remyelination in vivo. However, the effects of these approaches on OPC and oligodendrocyte populations have raised further questions about the influence of BDNF-TrkB on OPC proliferation, survival and origin of remyelinating cells outside of its well-defined role in myelin sheath formation. Astrocytes and neurons express high levels of TrkB isoforms and secrete BDNF [107,110] and could contribute to enhanced remyelination. Use of a structural BDNF-mimetic in mice with oligodendroglial TrkB deleted also resulted in an increase in myelination of small-diameter axons [89], strongly implying TrkB activation independent of oligodendroglia may have a role in promoting myelin repair that could be exploited for therapeutic benefit.

#### 3.2.3. Does TrkB Stimulated Remyelination Require Other Cell Types?

TrkB activation during remyelination in mice with oligodendroglial TrkB deleted still promoted remyelination of a subset of small diameter axons ranging 0.3–0.6 µm [89]. Other reports suggest that remyelination of small-diameter axons is activity-dependent [61,74], and it is possible neuronal TrkB may be involved. BDNF via neuronal TrkB activates glutamatergic neurotransmission [67,71,72], and blocking NMDA glutamate receptors in myelinating co-cultures has been seen to prevent the pro-myelinating effect of BDNF [61]. Whether this was dependent on blocking neuronal or OPC NMDA receptors was not discriminated. Importantly, it is not known if the specific effect of BDNF-TrkB on NMDA receptor activity is involved or required during myelin regeneration. Before speculating further, it is crucial to determine the roles of neuronal TrkB in remyelination through adopting an inducible conditional KO approach which selectively deletes TrkB in adult neurons without interfering in normal neural development.

The role of TrkB activation on astrocyte function in remyelination is contentious. In one study, TrkB stimulation of astrocytes in vitro with exogenous BDNF was observed to elevate nitric oxide production [111]. This was proposed to be a mechanism of neurodegeneration in EAE, which had an ameliorated disease course in mice in which astrocytic TrkB was deleted [111]. In contrast, targeting TrkB activation with TDP6 in the cuprizone model had no effect on reactive astrogliosis [89]. However, this does not consider that TrkB activation may alter astrocyte secretions [16], potentially enhancing remyelination by creating a pro-myelinating microenvironment. Further, astrocytes have high levels of the truncated TrkB isoform, which when activated by BDNF can evoke calcium signalling within astrocytes independently of glutamate release or neuronal activity [14]. Whether this aspect of BDNF-TrkB signalling has any influence on the remyelinating capacity of demyelinated CNS is untested. Potentially, it could influence either activity-dependent or -independent remyelination due to the effects of astrocytic calcium signalling on neuronal activity [112] and astrocyte secretion of BDNF itself [73]. These possibilities could be explored with ex vivo electrophysiology recordings on tissue slices from conditional KO mice with astrocytic BDNF secretion [110] or TrkB removed that had been subjected to de- and remyelination. 

## 4. Conclusion and Summary

BDNF has been proven to promote CNS myelination during development and to enhance remyelination following myelin injury. This is exciting as it suggests targeting BDNF signalling is a viable therapeutic strategy for neurologic conditions such as MS, stroke and traumatic injury where myelin damage is the prominent underlying cause of clinical dysfunction. However, use of exogenous BDNF as a therapy is problematic due to its suboptimal pharmaco-kinetic behaviour and complex receptor signalling. Future studies aimed at investigating the cellular and molecular mechanisms that execute the pro-myelinating effect of BDNF are required for maximising remyelinating outcomes. This includes investigating the potential role BDNF may have in activity-dependent myelin plasticity. Precise understanding of both the downstream pathways following oligodendroglial TrkB activation and their biological outcomes, as well as the contribution of neuronal or astrocytic TrkB to the influence BDNF exerts upon myelination, will provide novel insight into the myelinating process. Outcomes of these studies will ultimately lead us to identify new therapeutic approaches that effectively and specifically enhance myelin repair. Progress in the development of small molecule TrkB agonists offers hope that BDNF signalling can be harnessed for therapeutic benefit in human CNS demyelinating diseases. 

## Figures and Tables

**Figure 1 ijms-19-04131-f001:**
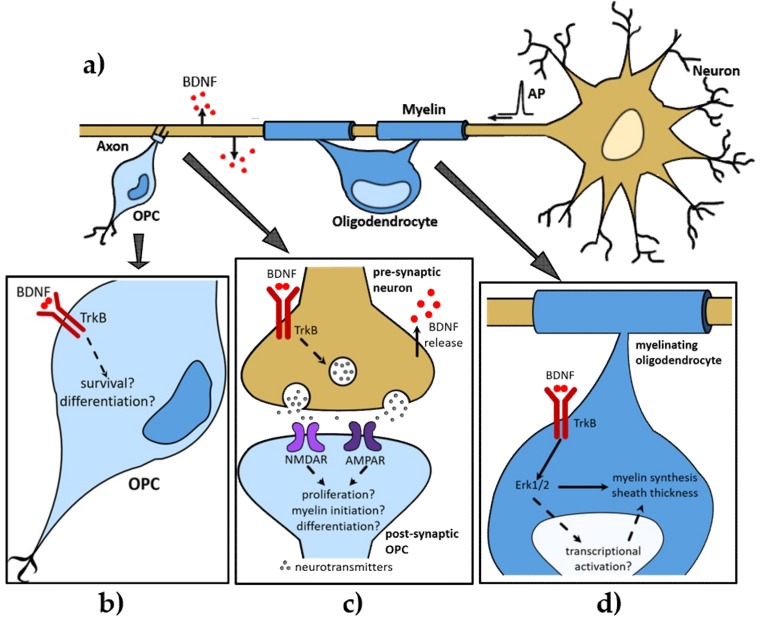
Schematic proposing the multi-faceted roles of brain-derived neurotrophic factor (BDNF)-tropomyosin-related kinase B (TrkB) signalling in central nervous system (CNS) myelination. (**a**) The myelinating process requires the oligodendrocyte progenitor cell (OPC) to contact the axon and differentiate into the mature oligodendrocyte that ensheaths the axon with myelin. Action potential (AP) firing by active neurons results in the release of BDNF along the axon. BDNF-TrkB signalling (**b**) could influence OPC survival and differentiation in development and after myelin injury and (**c**) promotes activity-dependent myelination by modulating glutamatergic (NMDAR; AMPAR) neurotransmission. It is well established that (**d**) BDNF-TrkB signalling *via* extracellular-related kinase 1/2 (Erk1/2) promotes the synthesis of myelin proteins and this influences myelin sheath thickness. Not known are the molecular mechanisms that underpin this effect and it is hypothesised that the BDNF-TrkB-Erk cascade results in transcriptional activation controlling myelin protein expression. Dashed arrows: hypothesised outcome of TrkB signalling; solid arrows: reported mechanism and outcome of TrkB signalling. See text for discussion.

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
