# Peer review of "Brain-Derived Neurotrophic Factor in Central Nervous System Myelination: A New Mechanism to Promote Myelin Plasticity and Repair"

_ijms, 2018, doi:10.3390/ijms19124131_

Reviewer 1 Report

This is an excellent review of the literature that indicates roles of BDNF on oligodendrocytes.  It is well written and clear and strongly argues that BDNF is an  important factor to consider when evaluating factors regulating  oligodendrocytes during development and in response to injury. One minor  correction is suggested with respect to the interpretation  of an early study to which the authors refer.   Although in vitro studies looking at the basal forebrain suggested that  BDNF increases numbers of oligodendrocytes, it was never shown that BDNF  affects survival there.  In fact, it was found that the number of MBP+ cells was increased  in response to BDNF treatment, while total cell numbers of  oligodendrocytes did not change, suggesting that BDNF affects  differentiation, rather than survival.  This possibility is consistent with subsequent in vivo studies that have been published.

Author Response

Point 1: One minor  correction is suggested with respect to the interpretation  of an early study to which the authors refer.   Although in vitro studies looking at the basal forebrain suggested that  BDNF increases numbers of oligodendrocytes, it was never shown that BDNF  affects survival there

Response 1: We thank the reviewer for this clarification and correcting our interpretation of this point. We have edited lines 102-103 to remove reference to any experimental evidence showing BDNF exerts a survival effect in vitro.

Reviewer 2 Report

I appreciated in the review the comprehensive approach, by which many explanation given in the literature and reported i the text are presented critically. On the other hand, the studies in this field and their presentation in this review, are mostly based on descriptive symptoms. What is missing, potentially interesting for the development of the studies, would be the suggestion of new, more molecular initiatives, helpful to identify the mechanisms sustaining the reported descriptive processes. Possibly this suggestion could be developed in the future.

Author Response

Point 1: What is missing, potentially interesting for the development of the studies, would be the suggestion of new, more molecular initiatives, helpful to identify the mechanisms sustaining the reported descriptive processes.

Response 1: We thank the reviewer for their critical feedback. We agree more molecular initiatives are required to identify the mechanisms that sustain the processes described in the review. This is evident in comments in lines 126-128, 314-319 and 369-370. Further, we also point out a couple of  targeted mechanistic studies that could probe the signaling mechanisms and biochemistry of the observed effects of BDNF on myelination, as evident in lines 184-186, 349-352 and 388-390. We view this as a key component that we identify and discuss in the review.

Reviewer 3 Report

This review elegantly summarises the latest development in BDNF-mediated mechanisms affecting CNS myelination. The manuscript is concise and well-focused on myelination in development as well as injury, and highlights the central role of TrkB and BDNF-mediated functional outcomes in demyelinating conditions. Clear consideration is given to the complexity of effects on the numerous cell types that express TrkB and/or release/bind BDNF. The axis of BDNF/TrkB/OPC is discussed in detail, including the shortcomings of current in vivo studies.The authors touched on the relevance of the numerous TrkB variants, so a very minor omission is the extrapolation of the TrkB isoform functional relevance in relation to BDNF signalling. Otherwise this is an exceptionally clear review with a well summarised and very effective figure and has my recommendation to publish as is.

Author Response

Response: We thank the reviewer for their comments.